# Ingestion of Okinawa Island Vegetables Increases IgA Levels and Prevents the Spread of Influenza RNA Viruses

**DOI:** 10.3390/nu13061773

**Published:** 2021-05-22

**Authors:** Kenji Gonda, Hideto Kanazawa, Goki Maeda, Chisa Matayoshi, Naoto Hirose, Yukiteru Katsumoto, Koji Kono, Seiichi Takenoshita

**Affiliations:** 1Department of Medicine, Daido Central Hospital, 1-1-37, Asato, Nah City 902-0067, Okinawa, Japan; opirineppu@gmail.com (H.K.); hime1116chisa614@icloud.com (C.M.); 2Department of Gastrointestinal Tract Surgery, Fukushima Medical University, 1, Hikarigaoka, Fukushima City 960-1295, Fukushima, Japan; kojikono@fmu.ac.jp; 3Fukushima Medical University, 1, Hikarigaoka, Fukushima City, Fukushima 960-1295, Japan; akiko-t@fmu.ac.jp; 4Department of Regional Agricultural System Section, Okinawa Prefectural Agricultural Research Center, 820, Makabe, Itoman City 901-0336, Okinawa, Japan; maedagok@pref.okinawa.lg.jp; 5Department of Research and Development section, Okinawa Industrial Technology Center, 12-2, Suzaki, Uruma City 904-2234, Okinawa, Japan; hirosent@pref.okinawa.lg.jp; 6Department of Faculty of Science, Fukuoka University, 8-19-1, Nanakuma, Jonan City 814-0180, Fukuoka, Japan; katsumoto@fukuoka-u.ac.jp

**Keywords:** Okinawan vegetables, influenza virus, upper respiratory tract infections, IgA, COVID-19

## Abstract

Background: It has been hypothesized that flavonoid ingestion stimulates immunity, promotes health, and prevents human illness. The aim of this analysis was to evaluate the association of the levels of immunoglobulin A (IgA) with the prevention of influenza infections and with the polyphenols contained in Okinawan vegetables. Methods: IgA, immunoglobulin G (IgG), immunoglobulin M (IgM), and soluble interleukin-2 receptor (sIL-2R) levels were measured in 44 outpatients who regularly ingested vegetables grown on Okinawa Island (200–300 g/day for ≥ 300 days/year) with no history of influenza infection and in 73 patients who ingested the vegetables irregularly or not at all with a history of influenza infection. Results: The patients who regularly ate Okinawan vegetables had higher IgA, IgG, and IgM levels than those who did not. On the other hand, patients who did not consume Okinawan vegetables and had influenza had lower IgA, IgG, and IgM levels. In addition, the IgA and IgG levels showed significant positive correlations with the sIL-2R levels in both groups. Conclusions: It may be beneficial to eat vegetables abundant in polyphenols every day. Secretory IgA antibodies are an important part of the immune defense against viral diseases. People who ingest Okinawan vegetables have high IgA levels and might be more likely to develop immunity against influenza RNA viruses.

## 1. Introduction

Our island, “Okinawa”, is in the southernmost prefecture of Japan also called Okinawa, which is composed of 160 islands scattered over a vast area of about 400 km north to south and 1000 km east to west. The island has natural and geographical characteristics that result in the classification of its climate as subtropical oceanic [1]. On our island, people have been eating a group of locally grown plants, called island vegetables, since ancient times. Our island is a crescent-shaped archipelago, and the natural forests hold a huge variety of unique subtropical plants with relatively high endemism [2]. The plants are known to produce phytochemicals [3]. The antioxidant activity of the phytochemicals in Okinawan vegetables, which grow under exposure to ultraviolet light stronger than that in other prefectures in our country, is high [4]. The UV treatments under different photoperiod regimes effectively raise flavonoids biosynthesis among all phenolic compounds and may upregulate chalcone synthase and flavonol synthase for flavonoid production [5]. The dietary flavonoids, especially their glycosides, are the most vital phytochemicals in the diet and are of great general interest due to their diverse bioactivity. Increasing scientific evidence has shown that flavonoids that are found in fruits and vegetables can have anti-inflammatory properties [6]. It has been suggested that flavonoids reduce the incidence of upper respiratory tract infections (URTIs) because they have a range of physiological effects in humans, including antiviral and anti-inflammatory effects [7]. Somerville et al. reported the essential role of flavonoids in the function of the respiratory immune system. Furthermore, their findings demonstrated that flavonoids decreased the incidence of URTIs with no adverse effects [8]. Okinawan vegetables are more likely to be rich in polyphenols (Table 1) (Figure 1 A–I). Polyphenols promote immunity to foreign pathogens. Different immune cells express multiple types of polyphenol receptors, which subsequently activate signaling pathways to initiate immune responses [9]. The early action of IgA humoral immune responses plays a critical role in protecting individuals. Polyphenols significantly increase IgA levels (Figure 1J) [10]. Secretory IgA antibodies are an important part of the airway mucosal immune system [11].

A major feature of our prefecture is that the influenza virus is prevalent year-round. There is no clear answer as to why influenza is prevalent on the island even in the summer. However, it is known that influenza epidemics occur year-round in tropical and subtropical regions, and Okinawa, which is the only prefecture with a subtropical location and climate, has an influenza pattern that is distinct from that in the rest of Japan. 

This causes the airway mucosa of island people to be constantly exposed to influenza antigens. Soluble interleukin-2 receptor (sIL-2R) levels have been reported to be increased in patients with IgA nephropathy [12], and interleukin (IL)-2 has been suggested to be involved in IgA production [13,14].

Given the food habits of the island peoples, it was assumed that there is a high rate of ingestion of flavonoids on the island. Therefore, IgA, IgG, and IgM levels were measured, assuming that the ingestion of these vegetables affects immunity. The associations of the IgA, IgG, and IgM levels of patients with the daily intakes of these vegetables were then examined. In addition, the potential differences in the IgA, IgG, IgM, and sIL-2R values of patients who ate these vegetables and had no history of influenza infection and those who did not eat them and had a history of influenza infection were also investigated.

The aim was to assess the mechanisms of immunoglobulin defense against the infection of the respiratory tract mucosa because the elimination of infection and the prevention of reinfection need to be addressed. Viral infection may be prevented, and its onset may be suppressed; therefore, immunity to influenza infection and coronavirus disease 2019 (COVID-19) may be acquired at an early stage by islanders.

## 2. Materials and Methods

### 2.1. Outcomes

#### 2.1.1. Primary Outcomes

The participants represented 153 of 392 outpatients, after excluding 239 outpatients who were vaccinated against influenza infection at least once in the year from August 2019 to September 2020. Of the 153 outpatients, 61 consumed 200–300 g/day of island vegetables for ≥ 300 days/year. Of these, 17 (11 men and six women) had influenza infections at least once during the period, and 44 patients (20 men and 24 women) did not have influenza infections within this period. Of the 153 patients, 92 consumed approximately 50 g/day of island vegetables for ≤ 100 days/year. Of these, 73 patients (42 males; 31 females) had influenza infections, and 19 patients (seven males; 12 females) did not have influenza infections.

#### 2.1.2. Secondary Outcomes

The IgA, IgG, and IgM levels of these patients were measured. The correlations among the IgA, IgG, and sIL-2R levels in the patients who ate vegetables and did not get influenza and among those who did not eat vegetables and got influenza were also investigated. Patients whose sIL-2R levels were measured did not show significant renal dysfunction. The IgA, IgG, IgM, and sIL-2R levels were measured in the blood of 153 patients without underlying systemic diseases (87 males, 66 females; average age, 69.4 years), including orthopedic patients (68) and patients with cerebrovascular disease and neurological disorders (85).

### 2.2. The Procedure for Extracting Phenolic Compounds from Vegetables of the Island

Phenolic compounds were extracted from freeze-dried vegetables twice (50 °C, 1 h) with methanol, and the solvent was evaporated. The total polyphenol content of the vegetable methanol extracts was determined using the Folin–Denis method [15] with minor modification. A phenol reagent (20 μL), an aqueous solution of 10% Na_2_CO_3_ (40 μL), and distilled water (120 μL) were added to 20 μL of the methanol extract, and the mixture was incubated at room temperature for 1 h. The absorbance was measured at 750 nm with a spectrophotometer. An aqueous solution of gallic acid was used as a standard [16]. The total amount of polyphenols (flavonoids) contained in the island vegetables was calculated using the gallic acid equivalence method (mg/100 g FW) (Table 1).

### 2.3. Statistical Analysis

IgA, IgG, and IgM levels were measured using the immunoturbidimetric method, in which the antibody reacts with an antigen to form an immune complex precipitate, the aggregate is irradiated with light, and the attenuation (absorbance) of the irradiated light due to scattering is automatically detected (SRL Inc., Tokyo, Japan). The serum concentrations of sIL-2R were determined using a chemiluminescent enzyme immunoassay (SRL Inc.). The cutoff values were as follows: IgA (110–410 mg/dL), IgG (870–1700 mg/dL), IgM (33–190 mg/dL), and sIL-2R (145–519 U/mL). Data are presented as means ± SD. The *p*-values were determined using Student’s t-test. Relationships between two variables were quantified using Spearman’s rank correlation coefficient. A *p*-value < 0.05 was considered significant. SAS software version 9.2 (SAS Institute Inc., Cary, NC, USA) was used for statistical analysis.

## 3. Results

### 3.1. Study Design

The participants represented 153 of 392 patients. Of the 153 outpatients, 61 consumed a high content of island vegetables in the year prior to recruitment, whereas 92 did not. Of the 61 patients who consumed a high content of island vegetables, 44 patients did not contract influenza during the year. On the other hand, of the 92 patients who did not consume a high content of island vegetables, 73 contracted influenza. The sample size was small and, thus, not sufficient to provide solid evidence. Thus, these findings are preliminary and should be supplemented by further studies on a larger number of subjects. A robust way to test the anti-influenza properties of Okinawan vegetables would be a controlled randomized clinical trial with measurable and clinically relevant outcomes (Figure 2). 

### 3.2. Patients Who Ate Island Vegetables vs. Those Who Did Not

Of the 61 outpatients who ingested 200–300 g/day of island vegetables for ≥ 300 days/year (total polyphenol amount: 100 ≤ mg/day), 44 did not contract influenza throughout the year, whereas the remaining 17 did. Of the 92 outpatients who ate less than approximately 50 g/day of island vegetables for ≤ 100 days/year (total polyphenol amount: ≤ 20 mg/day), 73 contracted influenza at least once. We also found that IgA (688.68 ± 85.50 and 512.11 ± 66.4 vs. 279 ± 67.97 mg/dL; *p* < 0.01and *p* < 0.05) (Figure 3A) and IgG (2238.40 ± 863.75 vs. 1295.52 ± 311.38 mg/dL) levels (Figure 3B) were significantly higher in the outpatients who ate high levels of island vegetables than those who did not. However, the IgG (1601.47 ± 539.30 vs. 1295.52 ± 311.38 mg/dL, *p* < 0.1) (Figure 3B) and IgM (97.00 ± 54.64 and 82.88 ± 40.39 vs. 70.05 ± 47.61 mg/dL; *p* = 0.13 and *p* < 0.1) (Figure 3C) levels were not significantly different between groups, despite their being higher in the outpatients who ate high levels of island vegetables than those who did not. 

### 3.3. Uninfected Patients Who Did Not Eat Island Vegetables

Of the 92 outpatients who did not ingested island vegetables, 19 did not contract with influenza throughout the year. Their IgA (424.57 ± 16.23 mg/dL) (Figure 3A), IgG (1566.60 ± 539.30 mg/dL) (Figure 3B), and IgM (75.94 ± 36.55 mg/dL) (Figure 3C) levels were within the reference ranges. 

### 3.4. IgA and IgG Levels Were Correlated with sIL-2R Levels

In patients with no history of influenza who ingested vegetables, the IgA (r = 0.57, *p* < 0.01) (Figure 3D) and IgG (r = 0.38, *p* < 0.05) (Figure 3E) levels were positively correlated with sIL-2R. The IgA (r = 0.30, *p* < 0.01) (Figure 3F) and IgG (r = 0.25, *p* < 0.05) (Figure 3G) levels in patients with influenza who did not eat island vegetables showed significant positive correlations with sIL-2R levels. The IgA and IgG levels of patients who had influenza who ate island vegetables and those who did not have influenza and did not eat island vegetables did not show any correlations with sIL-2R levels.

## 4. Discussion

### 4.1. IgA and Viruses

It has been reported that polyphenols significantly increase the amount of fecal mucin and IgA levels [10]. In addition, the induced IgA antibody is able to preferentially recognize a specific region of the viral antigen. It has a high neutralizing activity, and, upon viral infection, the IgA antibody multimer shows high protective activity, suggesting that the IgA antibody might then become an immune surrogate. The secretory IgA antibody, which is the most abundant antibody in the body, plays a leading role in the body’s defense against infectious diseases caused by viruses that target mucosal tissues. The content and context of immunoglobulin genes are altered during the differentiation of lymphocytes, and immunological diversity with somatic rearrangement is acquired in the body [17]. 

### 4.2. IgA, Polyphenols, and Immunity

Polyphenols can be used to regulate intestinal mucosal immune responses, allergic diseases, and antitumor immunity [9], and it is possible that the regular intake of polyphenols helps to maintain IgA production. Supplementation with buckwheat flavonoids also reportedly increases IgA levels. Additionally, it was confirmed that IgA and IgG are produced in higher quantities by the peripheral blood mononuclear cells of healthy people who consume wine polyphenols [18]. Polyphenols generally improve the Th1/Th2 cytokine balance, leading to cellular immunity dominating over humoral immunity and the subsequent inflammatory response that causes inflammation [19,20]. They also play a role in the release of cytokines required for cell differentiation. Reportedly, the acute respiratory failure caused by influenza and COVID-19 is due to the release of cytokines caused by cytokine storms. Thus, the IL-6/STAT3 pathway has been proposed as a therapeutic target for its prevention. Quercetin regulates the excessive secretory production of IL-4 and IL-6, which are inflammatory mediators, and improves the Th1/Th2 balance [21].

### 4.3. Why Okinawan Vegetables Are High in Polyphenols

Okinawa Island’s soil is composed of Ryukyu limestone made of ancient coral. Plants grown on coral land containing phenylalanine might have high flavonoid contents. The zooxanthellae produce an essential amino acid that cannot be synthesized in the bodies of corals; specifically, they synthesize phenylalanine and provide it to the coral [22]. Cinnamic acid, produced via the deamination of phenylalanine by the phenylalanine ammonia-lyase that is present in plants, is the starting material for the synthesis of a very diverse group of flavan-based flavonoids, which are present in most land plants. Almost all natural flavonoids exist as their *O*-glycoside or *C*-glycoside forms in plants [23]. A large-scale investigation of Okinawan rural people confirmed that those who consume more polyphenols in their diets have a lower risk of death [24]. Some patients in the present cohort ate 200–300 g/day of island vegetables for ≥ 300 days/year. The recommended intake of polyphenols is about 1000–1500 mg/day; however, the actual intake herein varied within 10–80% of the recommended value. The patients mainly ate the island vegetables described in Table 1.

### 4.4. Okinawan Vegetables High in Polyphenols

The getto or shell ginger plant (*Alpinia zerumbet*) (Figure 1A) contains phenolic acids (benzoic and cinnamic acid) [25] and apigenin, and it is said to promote longevity [26]. It is also believed to be effective for the treatment of human immunodeficiency virus (HIV) [27], which is a single-stranded RNA virus, similar to coronaviruses. *Saxifraga stolonifera* (Figure 1B) contains the flavonoid bergenin [28,29]. *Peucedanum japonicum* Thunb. (Figure 1C) contains the citrus flavonoids chlorogenic acid and rutin [30]. Interestingly, Suwa et al. showed that suitable soil conditions can increase the yield of polyphenols in *Peucedanum japonicum* Thunb. *Crepidiastrum lanceolatum* (Figure 1D) contains the flavonoid chicoric acid, which can inhibit the oxidation of low-density lipoprotein (LDL) cholesterol [31]. *Crepidiastrum laneolatum*, species located on the beach coast have smaller plant bodies but higher polyphenol contents than those grown in the fields; thus, they are picked on the beach for cooking. *Gynura bicolor* (Figure 1E) contains the flavonoids phenolic acid, carotenoid, and anthocyanin [32], which exhibit various absorption profiles in the 400–500 nm range because of long conjugated double bonds. The skin of *Benincasa hispida* (Figure 1F) contains quercetin [33,34]. Quercetin is believed to suppress the multiplication of influenza virus. *Momordica charantia* (Figure 1G), also called bitter melon, is a vegetable rich in polyphenols, and many studies of its antioxidant activity have been reported in our country and overseas [35]. The pulp of the ripe fruit and the whole unripe fruit show the highest amounts of total phenolic compounds [36]. Its antioxidant activity may be suitable for the treatment of patients with infection-related complications, such as those with severe influenza. *Citrus depressa* Hayata (Figure 1H) contains nobiletin [37]. Nobiletin, which is polymethoxylated, is present in the peel of Shiikuwasha; on the island, it is squeezed with the skin still attached and used as an ingredient. Nobiletin is known to promote the production of interferon-γ, a cytokine involved in the antiviral activity of NK cells [38]. *Ficus pumila L.* extracts contain rutin and apigenin (Figure 1I). The effect of *Ficus pumila L*. on metabolic syndrome depends on the extent of the high blood pressure, hyperlipidemia, and hyperuricemia occurring with the lifestyle diseases [39]. Human T-cell leukemia virus type 1 (HTLV-1) -infected patients who were administered *Ficus pumila L.* extract showed no HTLV-1-related symptoms [40].

### 4.5. Multimerized IgA Antibodies

Since island people are constantly at risk for influenza virus infection and URTIs, IgA levels were measured in a subset of the population, assuming that IgA antibody production is constant in that environment. The secretory IgA antibody, which takes the form of a large multimer, has higher virus-neutralizing activity than its dimeric form, and it also has neutralizing activity against antigenically distant viruses [41]. Patients consuming Okinawan vegetables had high IgA levels and were less susceptible to influenza infections. Although the influenza virus is known to evade immunity via the mutation of the antigens on its particle surface, the multimerized IgA antibodies distributed on the mucosa improve cross-reactivity, such that antibodies with a single variable region are reportedly able to cope with various viruses with different antigenicity (Figure 1J) [42]. IgA antibody production is constantly promoted by the consumption of Okinawan vegetables high in polyphenols. The binding between antigen and antibody involves intermolecular attractive forces, such as hydrophobic bonds, van der Waals forces, electrostatic forces, and hydrogen bonds; however, in the present study, the total binding force between the antigen and the antibody, known as the avidity, was not measured. IgG and IgM levels were also measured to confirm the immune response in vivo. It was found the IgM levels of patients with a history of influenza were not increased beyond the standard reference range. Simultaneously, IgA and IgG levels were also not high. 

### 4.6. IgA, IgG, and sIL-2R

Considering the possibility of high sIL-2R levels after influenza infection [43], sIL-2R levels were also measured in patients with a history of influenza infection who did not eat vegetables. IgA and IgG levels in patients with a history of influenza correlated with sIL-2R levels. IL-2 binds to IL-2R present on the cell surface and transduces a signal inducing the differentiation and proliferation of T cells, B cells, NK cells, monocytes, macrophages, etc. A relationship between sIL-2R and immunoglobulin in the small intestinal mucosa has been reported [44]. Increased levels of sIL-2R after influenza vaccination have also been reported [45]. The IgA and IgG levels of patients who ate island vegetables and did not have influenza and those of patients who did not eat these vegetables and had influenza both showed a positive correlation with sIL-2R levels. This means that the IL-2-mediated immune mechanisms are at least similar. In other words, it can be seen that the ingestion of island vegetables has the potential to increase IgA immunoreactivity and protect against infection by viruses other than influenza. This study evaluated a limited number of patients, and its results have regional specificity. In addition, the results do not clearly or theoretically explain the effect of the immune system on IgA antibody production from B cells and the rate of the internalization and absorption of polyphenols (Figure 1J) [46]. It may be beneficial to ingest vegetables abundant in polyphenols every day. 

### 4.7. The Role of Polyphenols in Defense against Coronaviruses

There is currently no effective medicine with which to prevent the spread of COVID-19 infection. Effective and safe pharmacological treatments need to be developed. The inhibition of nucleocapsid protein (N-protein) expression is the primary mechanism of polyphenols. Grape pomace extracts containing polyphenols interfere with nuclear protein expression, reduce viral replication, relieve the pathological complications of viral infection at the respiratory level, and regulate inflammation-promoting IL-6 in the airway mucosa. The antioxidant and anti-inflammatory properties of polyphenols have been established and may contribute to the treatment of persons with respiratory complications due to coronavirus infection [47]. 

## 5. Conclusions

In summary, secretory IgA antibodies are an important part of the immune defense against viral diseases. People who ingest island vegetables have high IgA levels and might be more likely to develop immunity against influenza. With the current prevalence of influenza and the COVID-19 pandemic, the development of antiviral drugs, the acquisition of collective immunity, symbiosis with viruses, and vaccination against RNA viruses are being evaluated to end the pandemic. Therefore, it may be possible to construct methods for preventing and limiting the spread of viral infections on the island, especially in the context of the current status of influenza and the COVID-19 pandemic. We hope that this report will help to promote different measures to combat future waves of influenza and COVID-19.

## Figures and Tables

**Figure 1 nutrients-13-01773-f001:**
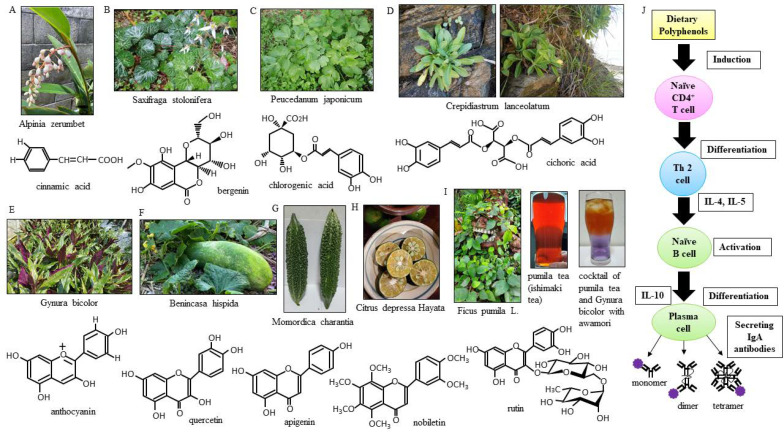
Polyphenol-rich island vegetables. The getto (*Alpinia zerumbet*) plant (**A**) is named “moon peach” because the flower buds have a peach-like shape, and it is called sannin on the island. It is also one of the flavors of Blue Seal Ice Cream. When written in our country’s Kanji characters, “yukinoshita” (*Saxifraga stolonifera*) (**B**) means “under the snow”, although it also grows on the island where it does not snow, and it is called minjaigusa on the island. *Peucedanum japonicum* (**C**) is called “peony windbreak” because its leaves are similar to peony flowers, and it is called sakuna on the island. *Crepidiastrum lanceolatum* (**D**) is called “hosobawadan” on the island. It was named as such because it has thinner leaves than *C. platyphyllum*. *Gynura bicolor* (**E**) has beautiful two-toned leaves that are green on the front and purple on the back, and it is sometimes called “island spinach” in English. *Benincasa hispida* (F) is a summer fruit, but it is called “wax gourd” because it can be stored as a whole (ball) until winter, and this island winter melon is called “shibui” and “watermelon” in English. *Momordica charantia* (**G**) is called “goya” on the island. The peculiar bitterness of goya is due to momordicin, which is contained in the skin. The level of vitamin C contained in the entrails inside goya is three times that contained in the skin. Goya has a conjugated linoleic acid that is contained in the seeds. *Citrus depressa* Hayata (**H**) is called “Shiikuwasha” on the island. Shiikuwasha is a wild mandarin orange from Okinawa. In Okinawan dialects, it is said that “shii” means sour and “kuwasha” means food. Before eating, Shiikuwasha is best when it is squeezed for juice and put on a dish or put in alcohol. *Ficus pumila L*. (**I**) is called “Ooitabi” on the island. During ancient times, the extracts of Ooitabi leaves were used for making Ishimaki tea in some areas of Okinawa prefecture. Island people drink about 200–300 mL of Ishimaki tea a day. Awamori is a distilled liquor made on the Ryukyu Islands, produced by saccharifying starch from the raw materials (rice with black koji, which is a rice malt) and then distilling the products. Rice malt contains polyphenols and increases IgA. Polyphenols are known to have various immunoregulatory effects (**J**). Naïve CD4+ T cells are activated in secondary lymphatic tissue, and they gain effector function and differentiate into Th2 cells. Th2 cells secrete IL-4 and IL-5, and they activate naïve B cells. Activated B cells are transformed into plasma cells that secrete antibodies. IL-10 secreted from activated B cells induces the production of immunoglobulins. IL-10 most strikingly induces the production of high amounts of IgA. Plasma cells release IgA. IL; interleukin.

**Figure 2 nutrients-13-01773-f002:**
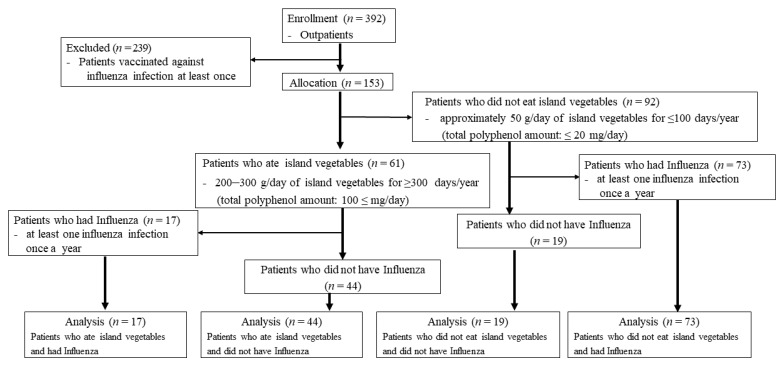
A participant flowchart and flow diagram for recruitment.

**Figure 3 nutrients-13-01773-f003:**
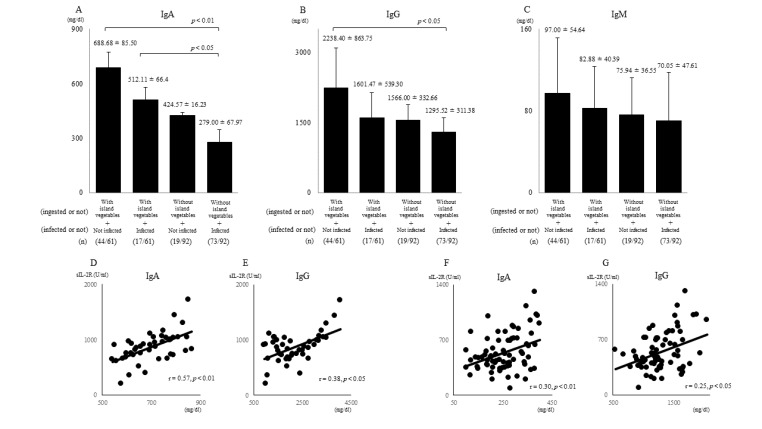
Immunoglobulin levels and relationships of sIL-2R levels with IgA and IgG levels. The IgA (**A**) and IgG (**B**) levels in the sera of the patients who ate Okinawan vegetables were significantly higher than those in patients who did not eat Okinawan vegetables (*p* < 0.01, *p* < 0.05, and *p* < 0.05). IgM (**C**) levels were higher in patients who ingested island vegetables. Among the patients who did not contract influenza throughout the year, the IgA (**D**) and IgG (**E**) levels of those who ingested island vegetables showed positive correlations with the sIL-2R levels (r = 0.57, *p* < 0.01, and r = 0.38, *p* < 0.05). Among the patients who contracted influenza at least once, the IgA (**F**) and IgG (**G**) levels of those who did not ingest island vegetables showed positive correlations with the sIL-2R levels (r = 0.30, *p* < 0.01, and r = 0.25, *p* < 0.05). (n): number of patients.

**Table 1 nutrients-13-01773-t001:** Polyphenol contents of the Okinawan vegetables.

Okinawan vegetables	Polyphenols (Flavonoids)	Content (mg/100 g FW) ^1^	Daily intake (mg/day)
Alpinia zerumbet	phenolic acids (benzoic and cinnamic acid)	150 (Leaf)	10–20
Saxifraga stolonifera	bergenin	10–100 (Leaf)	25–50
Peucedanum japonicum Thunb	chlorogenic acid	120–300 (Leaf)	10–180
		100–200 (Stem)	
		700–1300 (Flower)	
	rutin	70–200 (Leaf)	
Crepidiastrum lanceolatum	chicoric acid	144 (Leaf)	3.0–10
Gynura bicolor	phenolic acid	1428–1569 (Leaf)	70–200
	carotenoid	921–1007 (Leaf)	
	anthocyanin	2135–2407 (Leaf)	
Benincasa hispida	quercetin	4 (Fruit)	3.0–4.0
	rutin	12 (Fruit)	
Momordica charantia	phenolic acids (apigenin and chrysin)	964 (Ripe Fruit)	500–900
Citrus depressa Hayata	nobiletin	129–170 (dry-Peel)	10–20
Ficus pumila L.	rutin	1.0 (Leaf)	2.4
	apigenin	0.7(Leaf)	1.4

^1^ Equivalent to gallic acid per 100 g of fresh weight (mg/100 g FW). FW; fresh weight.

## Data Availability

Data described in the manuscript, code book, and analytic code will be made available upon request pending application and approval.

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
