# Peer review of "Ingestion of Okinawa Island Vegetables Increases IgA Levels and Prevents the Spread of Influenza RNA Viruses"

_nutrients, 2021, doi:10.3390/nu13061773_

Round 1
Reviewer 1 Report
The manuscript entitled "Ingestion of Okinawa island vegetables increases IgA levels and prevents the spread of influenza, RNA viruses" covers current topics.
The Introduction section describes the topic in detail, contains the necessary information for the general framework of the study and contains bibliographic references in line with the work content.
For the Materials and Methods section, what was written from lines 116 to 124 should be inserted before the statistical analysis paragraph and the analysis methods should be explained in more detail by also inserting the procedure for extracting phenolic compounds from vegetables of the island.
For the Results section, in the caption of Fig. 1 it would be necessary to insert in detail what the various graphs refer to and the statistical significance and eliminate the explanation of the results obtained.
The discussions are extensive, well described and supported by recent bibliography.
Reviewer 2 Report
The present study is interesting and appealing.
The sample size is pretty small and, maybe, not sufficient to provide solid evidence. This study should be indicated as a pilot or preliminary and should be followed by further studies on a larger number of subjects.
Results are reported in a confusional way. They should be grouped in single paragraphs.
Figures should be included in the text close to their first citation.
The CONSORT scheme should be reported previously in the results.
Information regarding polyphenols contained in Okinawa vegetables (both table and figure) should be moved into the introduction section.
The following published paper, reporting evidence about the role of polyphenols against coronaviruses, should be reported and quoted (10.3389/fmed.2020.00240)